# CHARTREF: BENCHMARKING FINE-GRAINED VISUAL ELEMENT LOCALIZATION IN CHARTS

## ABSTRACT

Humans interpret charts by first localizing visual elements—such as bars, markers, and segments—before reasoning over the data. In contrast, current multimodal models primarily rely on text reasoning, limiting their ability to leverage fine-grained visual information. To address this, we introduce CHARTREF, a dataset of 38,846 questions, answers, referential expressions, and bounding boxes across 1,141 figures and 11 chart types. Our key insight is that the chart-rendering code makes it possible to generate visual element localizations that are aligned with question–answer pairs. Given only the Python script, a large language model infers the semantics of plotted data, maps data series to visual encodings, and programmatically extracts bounding boxes, yielding visual annotations for charts at scale. Using CHARTREF, we benchmark multimodal LLMs and find 3–7% accuracy improvements on chart question answering when models are provided with ground-truth bounding boxes. We further evaluate vision and multimodal models on chart object detection and visual grounding, where models localize an expression referring to the data in the chart. While object detection exceeds 80 AP@50, visual grounding accuracy is only 2.8, revealing a significant gap: current models can recognize chart elements perceptually but struggle to integrate context cues from axes, legends, labels, and data to ground fine-grained textual references.

## 1 INTRODUCTION

When reading charts, humans localize relevant visual elements before extracting insights, grounding their reasoning in the visualized data. Building AI models with similar chart localization and reasoning capabilities propel applications in finance (Shu et al., 2025) and healthcare (Lee et al., 2024), support interactive systems where models communicate with humans through visual annotations (Hu et al., 2024), and advance document understanding (Ma et al., 2024) and scientific discovery (Lála et al., 2023; Yang et al., 2023b). Despite the importance of chart grounding, prior work has focused exclusively on evaluating models through question answering (Masry et al., 2022; Wang et al., 2024; Xia et al., 2025; Tang et al., 2025), which only measures the accuracy of textual answers and overlooks whether models actually localize and reason over the relevant visual evidence.

While vision-language models are proficient at localizing objects in natural scenes (Liu et al., 2024; Li et al., 2022; Zhong et al., 2022; Xiao et al., 2024; Fu et al., 2025), grounding phrases in charts requires fundamentally different visual reasoning skills, such as the ability to interpret elements such as legends and axis labels, understand coordinate systems, and navigate multi-panel subplot arrangements. For example, in Figure 1 (right), identifying the visual marker that corresponds to the phrase "average response for Control Group in session 5" requires understanding the correspondence between the legend and the markers, parsing the correct data series, and comparing the x-axis to the plotted data. This makes chart localization a process that current models have limited exposure to, as their pretraining focuses primarily on natural image-text alignment rather than supervision over chart-specific structures.

Additionally, unlike visual grounding for natural images, there are relatively few datasets for evaluating and enhancing chart visual grounding. Previous work has focused intensely on treating charts as a computer vision problem (Methani et al., 2020; Suri et al., 2025), applying off-the-shelf detection models to raw images to extract bounding box annotations. These approaches inherently limit

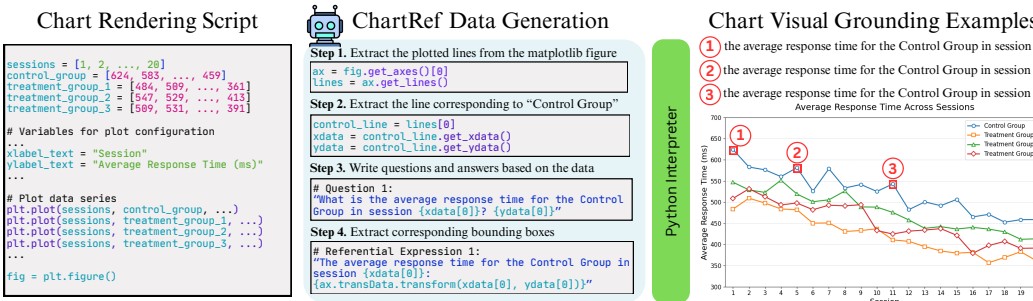

Figure 1: **CHARTREF dataset.** To generate a large-scale, diverse dataset of bounding box annotations paired with questions, answers, and referential expressions, we propose a data generation pipeline that takes as input the Python rendering script. We leverage the insight that the code representation contains information on the semantics of the plotted data, e.g. that `ax.get_lines()[0]` corresponds to the data for "Control Group", enabling generation of accurate questions and referential expressions that are grounded in contextual cues such as labels and ticks. Additionally, the code provides a means to extract bounding boxes for visual representations of the data by querying the underlying figure rendering. Our data generation results in CHARTREF, a benchmark for evaluating the chart visual grounding capabilities of multimodal models.

the diversity of the data to a select few chart types, such as bars, lines, and pie charts, to which standard object detection can be directly applied.

To address these shortcomings, we leverage the power of code as an intermediate medium to procedurally generate potentially unlimited source of chart images, visual element localizations, questions, and answers. This results in CHARTREF, a dataset of 38,846 paired examples across 11 figure types. Specifically, our data generation takes as input Python rendering scripts that encompass charts of diverse types and visual complexity. Our key insight is that without relying on the rendered image and given only the rendering script, a large language model can infer the semantic meaning of plotted data, associate data series with their corresponding visual encodings, and programmatically extract bounding boxes by querying the figure rendering as shown in Figure 1.

Using CHARTREF, we first evaluate multimodal foundation models on chart-question answering with and without the ground truth bounding boxes. In comparison to both standard prompting and chain-of-thought, models achieve a significant improvement in accuracy of 3-7% when given the ground truth annotation, motivating the development of models capable of fine-grained chart element localization. We next evaluate the capabilities of state-of-the-art vision and multimodal models on two tasks: 1) object detection, where the model detects all visual elements corresponding to the plotted data, and 2) visual grounding, where the model localizes an expression referring to a data point in the chart. Finetuning models on CHARTREF enhances performance significantly on both tasks – 10.6 AP@50 to 80.6 AP@50 for object detection and 0.3 Acc@1 to 2.8 Acc@1 for visual grounding. However, visual grounding performance is still far below that of object detection. These results highlight a critical gap in chart understanding and motivate future work on improving vision–language alignment to enable fine-grained chart visual grounding.

## 2 CHARTREF DATA CURATION

In Section 2.1, we first detail our pipeline for generating chart questions and answers with corresponding referential expressions and bounding boxes for the visual elements. In Section 2.2, we describe our procedure for postprocessing the generated questions and answers to facilitate evaluating numerical answers and eliminate ambiguous questions with a multimodal LLM verifier. In Section 2.3, we detail key statistics of CHARTREF.

### 2.1 DATA GENERATION

Given a matplotlib Python script that renders a chart, our data generation pipeline uses an LLM to synthesize code that extracts bounding boxes from the rendered visualization. This approach

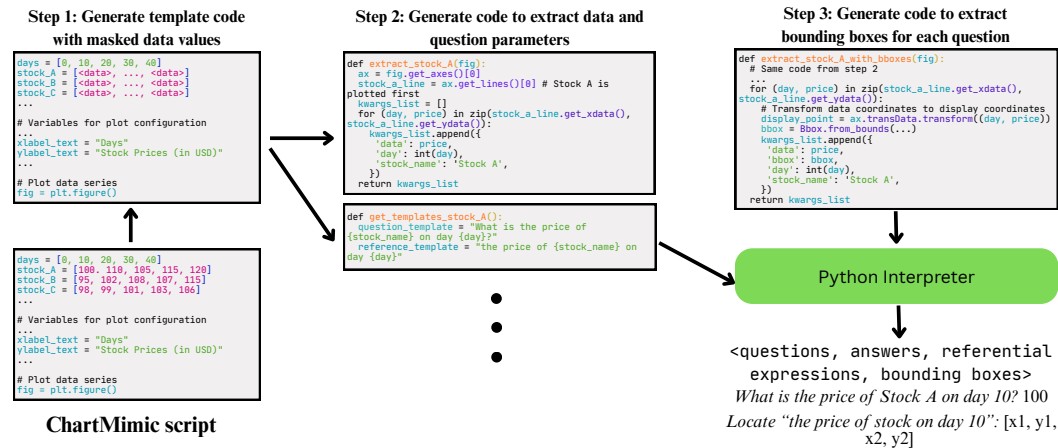

Figure 2: **Data generation pipeline.** Given the Python script from ChartMimic, we generate paired `<question, answer, referential expression, bounding box>` using the following data generation pipeline. In Step 1, we prompt a LLM to mask out the plotted data values, while preserving information required to interpret the chart, such as labels and coordinates at which the data points appear. In Step 2, the model identifies all data series that are masked out and writes code that extracts the data values and identifying parameters, as well as the question and referential expression templates. Applying the identifying parameters to the templates results in the questions and referential expressions that align with the data values. In Step 3, the model modifies the data extraction code from Step 2 to additionally extract the bounding boxes.

is necessary because unlike SVG or HTML formats where spatial coordinates are directly embedded in the markup, matplotlib scripts do not explicitly expose this spatial information. Instead, the figure object (e.g., `figure = plt.figure(...)`), stores both the ground truth data and the coordinates of the visualized data in image space. To extract both the underlying data and their bounding boxes, an LLM generates code that programmatically queries the figure object's properties. For example, the axes can be extracted with `figure.get_axes()`, and all line objects can be further collected with `ax.get_lines()` for each axis. This allows access to the data points via `line.get_xdata()` and `line.get_ydata()`, which can be converted to pixel coordinates using `ax.transData.transform()`.

Concretely, our data generation pipeline is as follows. As shown in Figure 2, we first prompt an LLM to generate template versions of a given Python script, where the plotted data is masked out but all other code is unchanged. This step ensures that the model writes code that extracts data directly from the figure object rather than hardcoding values, while still providing sufficient context—labels, axis titles, and visualization parameters—to infer the correspondence between data values and their textual descriptions. In the next stage, the LLM is prompted with the template to first identify all data series that are plotted by the code and for each data series, the LLM synthesizes a function that extracts the visualized data and parameters specifying how each series is visualized. These parameters are then used to instantiate question and referential expression templates. For example, if a data series represents "Stock A" values over a set of days, the generated questions correspond to individual data values, such as "What is the value of Stock A at day 10?", with the referential expression highlighting the corresponding visual element in the chart. After the data is extracted, the model is prompted to augment the data extraction code to additionally extract the bounding box for the visual element that corresponds to the visual element. When there are no explicit markers, we instruct the model to generate a $10 \times 10$ bounding box. See Fig 3 (top) for an overview of the types of questions, answers, and bounding boxes generated for each chart type. We use Claude Sonnet 4 (Anthropic, 2025) for all stages of the pipeline. See Appendix B for the data generation prompts and Appendix E for examples of the data in CHARTREF.

We organize the generated data into two tasks: chart question answering and chart element localization as shown in Figure 3 (bottom). To assess chart question answering with and without bounding boxes, we use the generated questions, answers, and bounding boxes. For chart element localization,

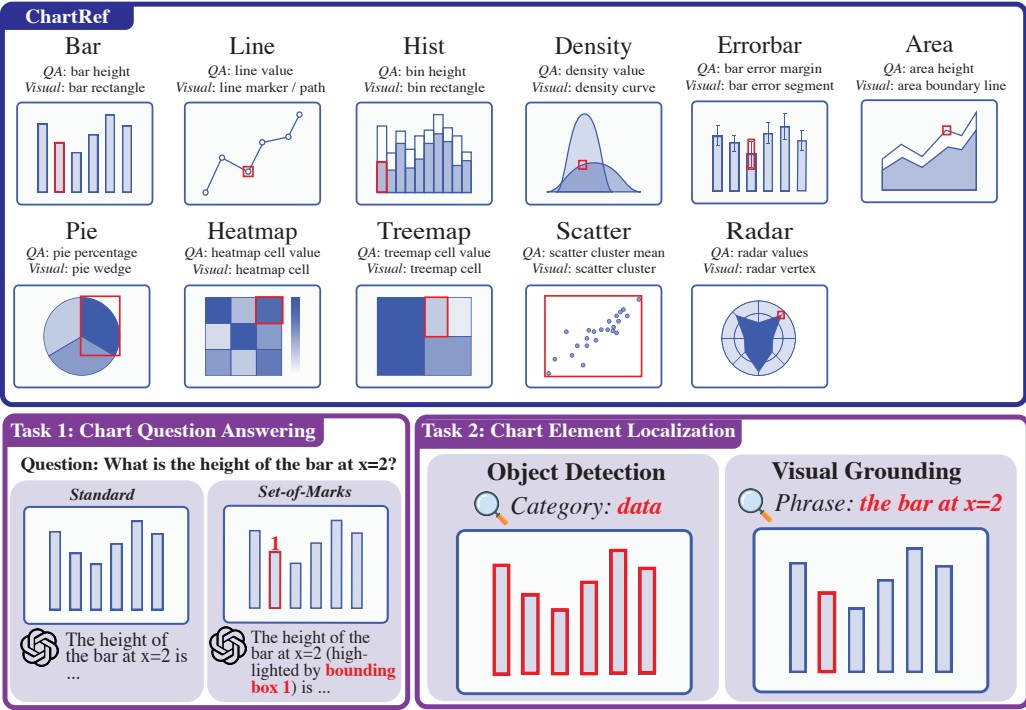

Figure 3: **CHARTREF overview. Top:** CHARTREF consists of figures across 11 chart types, with questions that require correctly extracting the underlying data and are paired with the corresponding answers, referential expressions, and bounding boxes around individual visual elements. For example, questions for bar plots ask about values that are visualized by the bar rectangle. **Bottom Left:** CHARTREF allows us to investigate the capabilities of multimodal foundation models on chart question answering, with both the original image and annotated with the ground truth bounding box. **Bottom Right:** With the ground truth bounding boxes, we benchmark models on two vision tasks: object detection, where all visual elements representing the data are identified, and visual grounding, where given a phrase, the model localizes a specific data element.

we consider two vision tasks: object detection and visual grounding. For object detection, we use all annotated bounding boxes for a given figure as ground truth. For visual grounding, by the nature of our data generation, there is a one-to-one mapping between referential expressions and bounding boxes.

## 2.2 DATA POSTPROCESSING

After generating (question, answer, referential expression, bounding box) tuples, we post-process the questions and answers as follows. We employ a multimodal LLM to determine the appropriate precision level for each ground truth answer and establish an error margin based on the spacing of relevant axis ticks. For numerical answers, we consider a prediction correct if $|\text{round}(y_g, p) - y_p| \leq \epsilon$, where $y_g$ is the ground truth, $y_p$ is the predicted answer, $p$ is the determined precision level, and $\epsilon$ is the error margin. For text answers, we use exact string matching.

We then validate the quality of the questions using a multimodal LLM to classify each question into one of three categories: 1) ambiguous questions, where there is not enough context in the chart to answer the question, 2) defective questions, where the answer is trivially answered without looking

Table 1: **Data generation statistics.**

| Statistic | Value |
|---|---|
| *Total* | |
| # of Examples | 44345 |
| # of Figures | 1259 |
| # of Chart Types | 11 |
| | |
| *Object Detection* | |
| # of Annotations | 40336 |
| # of Figures | 1148 |
| | |
| *Visual Grounding* | |
| # of Examples | 38846 |
| # of Figures | 1141 |

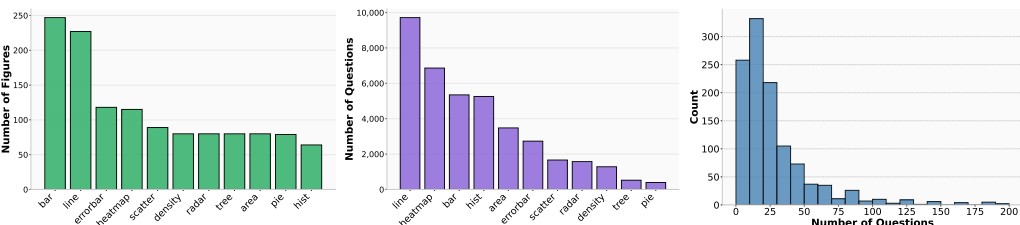

Figure 4: **CHARTREF statistics. Left:** The number of figures per type shows that the dataset is balanced across figure types. **Middle:** The number of questions per type reflects that certain figure types tend to reflect much denser information – for example a pie chart with several wedges will have fewer examples than a line plot reflecting tens or hundreds of individual examples. **Right:** The distribution of the number of questions show that the plots are densely annotated with bounding boxes, with many figures having over 20 paired examples.

at the chart, and 3) valid questions that do not fall into the above two categories. This ensures that the resulting questions do not reference information that is not available in the chart image. See Table 1 for the statistics. We use Claude Sonnet 4 as the multimodal LLM and detail the postprocessing prompts in Appendix C.

## 2.3 DATASET STATISTICS

We apply our pipeline to ChartMimic (Yang et al., 2025), a diverse dataset of human-curated Python chart rendering scripts with 2400 total figures across 22 different chart types. We select 11 of the chart types for which we could generate accurate bounding box annotations. In total, we generated 44345 paired question, answer, referential expression, and bounding box from 1259 figures. For object detection, we filtered out bounding box annotations with greater than 0.95 IoU overlap and did not include density plots, which represent continuous functions or distributions rather than discrete points, resulting in 40336 bounding boxes across 1148 figures. For visual grounding, we filtered out ambiguous and defective questions as discussed in Section 2.2, in addition to duplicate referential expressions. This resulted in a total of 38846 filtered examples across 1141 figures. In Figure 4, we analyze the distribution of figures per type, showing that although there is a higher proportion of bar and line plots, our data is balanced. The number of questions per figure type reflects the information density of plots – for example, line plots contain many individual data points. The distribution of the number of questions shows that the figures are densely annotated, with many having over 20 paired examples.

## 3 RELATED WORK

We outline prior work in assessing chart question answering, which have paired (chart image, question, answer) but lack visual annotations, as well as work that curates visual annotations for charts and other structured visuals, but often are limited to a subset of simple chart types, lack alignment between annotations and question-answers, or require human annotators.

**Chart Question Answering Datasets.** One line of work (Masry et al., 2022; Xu et al., 2024; Wang et al., 2024; Masry et al., 2024; 2025; Xia et al., 2025; Tang et al., 2025) focuses on evaluating chart question answering (CQA) that involve visual and logical reasoning over charts. Some benchmarks are human annotated – for example, ChartQA (Masry et al., 2022) consists of bar, line, and pie charts collected through web-crawling and human-annotated questions and answers. ChartQAPro (Masry et al., 2025) has been proposed as a CQA dataset with enhanced visual diversity. CharXiv (Wang et al., 2024) sources its charts from arXiv papers and evaluates multimodal LLMs on descriptive questions on extracting information from basic chart elements and reasoning questions requiring synthesizing information across multiple visual elements. ChartMuseum (Tang et al., 2025) contains expert-annotated questions focusing on visual reasoning that is difficult to perform with textual chain-of-thought. Other benchmarks are curated in an automated manner, involving multimodal LLMs in the loop. ChartX (Xia et al., 2025) uses GPT-4 to automatically generate chart title and

types that are aligned with CSV data for chart perception questions and task templates to generate cognition questions that builds on the perception questions. ChartBench (Xu et al., 2024) first generates the chart using LLMs to synthesize chart themes and JSON data and subsequently synthesizes questions and answers from chart templates. ChartGemma (Masry et al., 2024) proposes a data curation pipeline that leverages a multimodal LLM to directly generate questions from chart images.

**Visual Annotation Datasets.** Prior work has collected bounding box annotations for chart elements but typically only for a narrow range of chart types. FigureQA (Kahou et al., 2018) generates bounding box annotations from Bokeh (de Ven, 2018) for bar, lines, and pie charts, but these annotations are limited in visual diversity and lack alignment between annotations and questions. Other works leverage off-the-shelf vision models, such as PlotQA (Methani et al., 2020), which uses Faster-RCNN (Ren et al., 2017) to detect all bars, lines, and chart interpretation elements such as titles and labels. ChartLens (Suri et al., 2025) uses instance segmentation to annotate bar and pie charts and a specialized Transformer model to detect lines. Because these approaches depend on computer vision models for localization, they are inherently constrained to chart types where such models perform reliably. Orthogonal to our work, prior works (Battle et al., 2018; Zhu et al., 2025a) have developed pipelines for extracting annotations from SVG scripts, which have spatial coordinates directly embedded in the source code. Beagle (Battle et al., 2018) extracts circles, rectangles, line, and paths from charts. OrionBench (Zhu et al., 2025a) focuses on automatically synthesizing annotations for chart and human-readable objects embedded in infographics. Recently, RADAR (Rani et al., 2025) has curated human-annotated bounding boxes that correspond to individual CQA reasoning steps.

## 4 EXPERIMENTS

### 4.1 DO VISUAL ANNOTATIONS ENHANCE CHART PERCEPTION?

To motivate the use of bounding boxes to locate relevant visual elements for chart question answering, we evaluate the performance of multimodal LLMs on our synthesized dataset, where questions and answers are paired with ground truth bounding boxes that localizes the visual element corresponding to the answer. This enables us to evaluate the impact of annotating the ground truth bounding box on chart perception. We evaluate three closed source models, GPT-5 (OpenAI, 2025b), GPT-o3 (OpenAI, 2025a), Gemini-2.5-Pro (Comanici et al., 2025), and

Table 2: **Set-of-Marks with bounding boxes enhances chart perception.**

| Model | Direct | CoT | SoM |
|---|---|---|---|
| Qwen2.5-VL-72B | 67.6 | 66.6 | **70.2** |
| InternVL3-78B | 67.9 | 66.5 | **69.6** |
| GPT-5 | 80.0 | 80.0 | **83.8** |
| GPT-o3 | 78.4 | 76.8 | **81.4** |
| Gemini-2.5-Pro | 72.9 | - | **79.7** |

two open source models, Qwen2.5-VL (Bai et al., 2025), and InternVL3 (Zhu et al., 2025b). For all models, we experiment with three settings: standard prompting, chain-of-thought, and set-of-marks (SoM) prompting (Yang et al., 2023a), a visual prompting technique designed to ground model responses in visual cues. In our SoM prompting, the ground truth bounding box is overlayed on the image with a numerical label, allowing models to reference it in their answers. Results are shown in Table 2.

In comparison to standard prompting, set-of-marks prompting enhances performance by 2-3% for open-source models and 3-7% for closed-source models. In contrast, chain-of-thought prompting does not improve performance and even degrades it for Qwen2.5-VL, InternVL3, and GPT-o3. These results demonstrate that when the relevant chart visual elements are localized, downstream question answering is improved as the model is able to ground its reasoning in the bounding boxes.

### 4.2 OBJECT DETECTION

Using the bounding boxes, we evaluate vision models on object detection, where the task is to detect all visual elements corresponding to discrete data elements. We consider two types of models: traditional object detection models that must be trained to evaluate new categories, as well as foundation models capable of zero-shot detection. For traditional object detection models, we evaluate YOLOv3 (Redmon & Farhadi, 2018), Faster-RCNN (Ren et al., 2017), RTMDet (Lyu et al., 2022),

Table 3: **Object detection performance. Left:** Traditional object detection and foundation models finetuned on CHARTREF. **Right:** Zero-shot inference with foundation models.

| Model | AP | $AP_{50}$ | $AP_{75}$ | AR |
|---|---|---|---|---|
| YOLOv3 | 25.0 | 52.8 | 20.4 | 37.7 |
| Faster-RCNN | 54.1 | 60.4 | 60.4 | 67.4 |
| Co-DETR | 64.0 | 79.8 | **69.7** | 74.8 |
| RTMDet | 38.0 | 56.2 | 40.1 | 16.0 |
| MM-GD | 61.5 | 79.1 | 67.6 | 72.8 |
| LLMDet | **64.2** | **80.6** | 68.5 | **75.1** |

| Model | AP | $AP_{50}$ | $AP_{75}$ | AR |
|---|---|---|---|---|
| GroundingDINO (T) | 5.2 | 8.7 | 5.1 | 20.3 |
| GroundingDINO (B) | 4.7 | 7.8 | 4.7 | 20.2 |
| GLIP (T) | 3.8 | 6.6 | 3.8 | 13.2 |
| GLIP (L) | 5.8 | 8.7 | **5.8** | 16.3 |
| MM-GD T) | 4.1 | 8.0 | 3.6 | 26.4 |
| MM-GD (B) | 5.8 | 10.6 | 5.3 | 27.1 |
| MM-GD (L) | 3.6 | 6.2 | 3.7 | 26.7 |
| LLMDet (T) | 5.8 | 10.6 | 5.3 | 28.2 |
| LLMDet (B) | **6.3** | **12.2** | 5.6 | **30.0** |
| LLMDet (L) | 3.3 | 6.6 | 2.9 | 23.8 |

and CoDeTR (Zong et al., 2023). For vision models, we evaluate Grounding DINO (Liu et al., 2024), MM-Grounding-DINO (MM-GD) (Zhao et al., 2024), GLIP (Li et al., 2022), and LLMDet (Fu et al., 2025). As these models are not adapted to charts, we evaluate both the pretrained and finetuned models on our dataset.

**Results and Analysis.** Results for zero-shot inference and finetuned models on CHARTREF are shown in Table 3. For zero-shot inference, we benchmark vision foundation models across different sizes and show that the best performance achieved is 6.3 AP and 30.0 AR. Notably, increasing model size does not always enhance performance, demonstrating the out-of-domain shift from natural images to chart object detection. After finetuning on CHARTREF, AP improves from 6.3 to 64.2 and AR improves from 30.0 for 80.6 for LLMDet, with similar improvements seen in MM-GD. The best traditional object detection model, Co-DETR, achieves 64.0 AP and 74.8 AR.

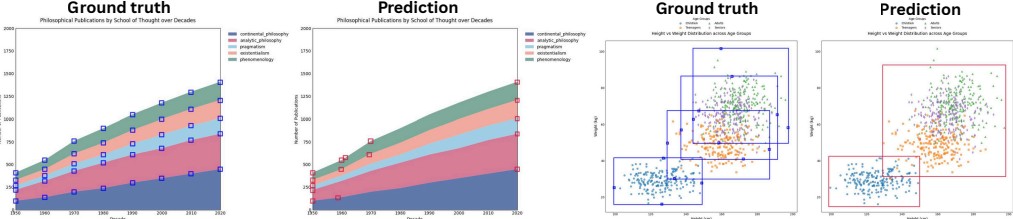

Figure 5: **Object Detection Example Errors.** We visualize errors made by Co-DETR, with ground truth bounding boxes displayed in blue and predicted bounding boxes displayed in red. **Left:** For area charts, data is plotted at each x-tick. However, the predictions are not able to capture the correct data points and are missing bounding boxes where the slope of the lines are constant and lack salient features. **Right:** For scatter plots, the ground truth visual elements include clusters of data points. Although the predictions are able to capture the blue cluster as it is well-separated from the other data points, the model does not correctly separate the overlapping orange, purple, and green clusters.

Additionally, we conduct a qualitative analysis in Figure 5 of Co-DETR's errors. We find that the models have an overreliance on salient perceptual features. For example, on the left chart of Fig. 5, the model struggles to identify individual data points at the x-ticks. On the right, Co-DETR does not correctly separate overlapping clusters that denote separate data series.

We further analyze Co-DETR's performance across different chart types in Table 4. We find that the lowest performance is for line and scatter plots, as the bounding boxes around individual markers in the line or scatter plot are typically small, whereas the model performs best for treemaps, pie charts, and bars, which contain fewer fine-grained bounding boxes and are represented by well-defined shapes, e.g. rectangles for treemaps and bars and pie wedges for bar charts.

Table 4: **Object detection performance across chart types.** Both AP@50 and AR scores are above 0.8 for errorbar, bar, tree, pie, and heatmap plots. The data for these plot types are represented by larger geometric shapes, such as rectangles, wedges, and grid cells, making it easier for the model to detect. In contrast, detecting individual data points for line and scatter plots are more difficult, as these visual elements are more fine-grained.

| Metric | Tree | Bar | Errorbar | Line | Radar | Hist | Area | Scatter | Heatmap | Pie |
|---|---|---|---|---|---|---|---|---|---|---|
| $AP_{50}$ | 92.3 | 91.3 | 80.0 | 60.4 | 74.3 | 84.8 | 85.6 | 69.4 | **95.6** | 82.4 |
| AR | **99.0** | 96.4 | 86.7 | 50.5 | 65.8 | 78.2 | 65.5 | 69.8 | 92.7 | 98.4 |

## 4.3 VISUAL GROUNDING

We benchmark both vision models and multimodal LLMs on chart visual grounding, where given a phrase, such as "the bar for Model A", the model outputs the correct bounding box that localizes the phrase. For vision models, we consider the same vision foundation models discussed in Section 4.2. We evaluate both zero-shot capabilities as well as finetune models on our CHARTREF. For reference, we additionally benchmark multimodal LLMs with zero-shot referential capabilities, including GPT-4o, Qwen2.5-VL, and InternVL3. We show the results in Table 5.

**Results and Analysis.** Among zero-shot vision models, performance is low, with all models achieving below 1 Acc@1. Larger vision models do not offer significant performance improvements, demonstrating that current models are not well-adapted to the task.

Table 5: **Visual grounding performance**

| Model | Acc@1 | Acc@5 | Acc@10 |
|---|---|---|---|
| *Zero-shot* | | | |
| GroundingDINO (T) | 0.6 | 2.8 | 4.9 |
| GroundingDINO (B) | 0.7 | 3.0 | 5.3 |
| GLIP (T) | 0.4 | 0.5 | 0.5 |
| GLIP (L) | 0.5 | 0.5 | 0.6 |
| MM-GroundingDino (T) | 0.3 | 2.7 | 5.3 |
| MM-GroundingDino (B) | 0.2 | 2.8 | 5.3 |
| MM-GroundingDino (L) | 0.3 | 3.1 | 5.9 |
| LLMDet (T) | 0.5 | 3.9 | 7.4 |
| LLMDet (B) | 0.6 | 3.6 | 6.9 |
| LLMDet (L) | 0.6 | 3.3 | 5.7 |
| *Finetuned* | | | |
| MM-GroundingDino (T) | **2.8** | **13.5** | **22.9** |
| LLMDet (T) | 2.3 | 12.0 | 21.3 |
| GPT-4o | 0.7 | - | - |
| Qwen-2.5-VL-72B | 0.5 | - | - |
| InternVL3-78B | 0.2 | - | - |

In comparison, multimodal LLMs perform similarly to the zero-shot foundation models, in spite of the increased amount of training on diverse image sources, which often include chart data, and on chart question-answering. Finetuning vision models yields a significant improvement: MM-GroundingDINO improves from 0.3 to 2.8 and LLMDet's accuracy improves from 0.6 to 2.3. However, the performance on visual grounding is still far below the object detection capabilities, which are close to perfect. This suggests that even though the vision backbone can be adapted to detect visual elements in charts, both vision foundation models and multimodal LLMs lack the vision-language alignment to understand the semantic information of charts beyond perceptual features.

For MM-GroundingDINO and GPT-4o, we visualize the visual grounding results in Figure 6. We find that MM-GroundingDINO tends to predict bounding boxes that do correspond to a visual element in the plot; however, it is not able to correctly identify the visual element referred to by the text. In contrast, GPT-4o's predictions tend to be imprecise, even in the presence of visual markers and text. These findings highlight that the main bottleneck lies in linking text references to the correct visual elements: vision foundation models can localize objects but fail to disambiguate which element is referenced, while multimodal LLMs struggles with precise localization altogether. Closing this gap will require improving models' ability to jointly reason over visual structure and textual cues, rather than relying solely on perceptual or linguistic features.

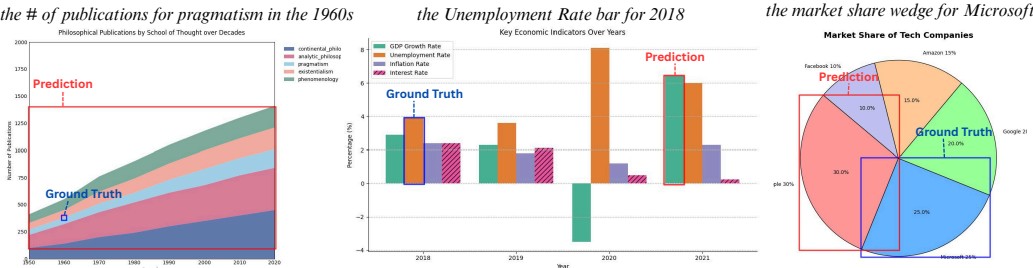

(a) Examples of MM-GroundingDINO visual grounding errors.

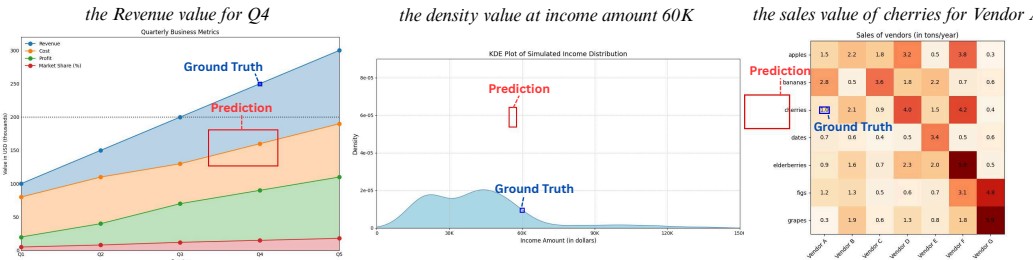

(b) Examples of GPT-4o visual grounding errors.

Figure 6: **Visual grounding errors.** Although MM-GroundingDINO tends to produce bounding boxes that are more closely aligned to the visual elements in the plot, it is not able to localize the correct visual element. **Top Left:** MM-GroundingDINO predicts a bounding box that contains multiple data points. **Top Middle:** The predicted bar does not match ground truth. **Top Right.** Even for a plot that represents only 6 distinct data points, MM-GroundingDINO localizes the wrong pie wedge. In contrast, GPT-4o's predictions often do not precisely localize the prediction. **Bottom Left:** The predicted bounding box is vertically offset from the ground truth and is also much larger than the visualized markers. **Bottom Middle:** GPT-4o's prediction is vertically offset and localizes an empty region of the plot. **Bottom Right:** The prediction is horizontally offset from the ground truth.

## 5 CONCLUSION

In this work, we present CHARTREF, a large scale dataset of 38846 paired (chart question, answer, referential expression, and bounding box) across 11 figure types. Our data curation pipeline leverages the chart's Python rendering code to programatically extract bounding boxes of visual elements that are aligned with questions and answers. With CHARTREF, we demonstrate that access to ground-truth bounding boxes improves chart question answering, motivating approaches capable of chart visual grounding. We thus benchmark vision models and multimodal LLMs on chart object detection and chart visual grounding. Through finetuning on CHARTREF, vision models can be adapted to detect all visual elements corresponding to the underlying data. However, both vision models and multimodal LLMs struggle to achieve comparable visual grounding performance when given a referential phrase. This gap highlights that chart visual grounding requires novel advances in text-vision alignment that would allow models to integrate diverse contextual cues, such as the legend, axis ticks, and subplot arrangement, to localize fine-grained information in charts. CHARTREF serves as a tool for both evaluating and training models on chart visual grounding, inspiring future work in models capable of human-like visual grounding.

**Limitations.** Because CHARTREF is generated from Python rendering code, it may not fully capture the stylistic variability, noise, or imperfections of real-world charts, potentially limiting model generalization to figures from research papers, reports, or scanned images. Additionally, the referential expressions in CHARTREF only require extracting individual data from the chart and do not cover more complex structures, such as comparisons or relationships between multiple data points. Finally, while the dataset spans 11 chart types, it does not encompass all visualization formats, including network diagrams, 3D plots, or interactive charts.

## 6 ETHICS STATEMENT

Our contribution represents a step towards developing models capable of localizing relevant visual elements in charts, which enhances the interpretability of model outputs and supports more transparent AI systems. This advancement contributes to society by enabling better understanding and validation of automated chart analysis, ultimately benefiting applications in data visualization, accessibility, and decision-making processes.

We have carefully reviewed all applicable ethical guidelines and believe our work adheres to the principles of scientific excellence, transparency, and responsible research conduct. Our dataset generation uses ChartMimic as input, which is available under the Apache 2.0 license, and we will adhere to the terms of this license when releasing CHARTREF, thus respecting intellectual property, privacy, and confidentiality guidelines.

## 7 REPRODUCIBILITY STATEMENT

To ensure reproducibility of our work, we provide comprehensive details of our methodology and experimental setup. The main text includes a detailed overview of our data generation pipeline, explaining the key steps for creating CHARTREF. Appendix B contains all prompts used in our LLM-based data curation process, enabling the replication of our dataset generation approach. Additionally, we detail experimental settings, hyperparameters, and evaluation protocols in AppendixD. We plan to release CHARTREF and code upon publication to facilitate further research and reproduction of our results.

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

## A   USE OF LARGE LANGUAGE MODELS (LLMS)

Large Language Models (LLMs) were used to improve the phrasing and clarity of the paper, particularly in the introduction and abstract, where we asked the model to improve an existing draft. LLMs were also used to enhance the prompts used in the data curation and verification pipeline. They did not play a significant role in research ideation.

## B   DATA GENERATION PROMPTS

Below is the prompt for generating templates from the input Python script.

Stage 1: Template Generation Prompt

```
You are a helpful assistant. Given a python script as follows:
```python
{code}
```

Your task is to replace the data that is plotted to placeholder <data>
tokens.

## Replace with <data>:
- Data values that appear as visual elements within the plot, such as
bars, lines, points, etc.
- Text labels that appear INSIDE the plotted area (annotations, data
labels on visual elements)

## Keep unchanged:
- Axis labels, titles, legends (all legend text)
- Tick labels and categorical labels
- All plotting parameters (colors, sizes, styles, limits)

Example:
```python
# Original
data = [10, 25, 30]
labels = ['A', 'B', 'C']
plt.bar(labels, data)
plt.title('Chart')

# Template
data = [<data>, <data>, <data>]
labels = ['A', 'B', 'C']  # Keep unchanged
plt.bar(labels, data)
plt.title('Chart')  # Keep unchanged
```
```

Below is the prompt for extracting individual data elements and identifying arguments, as well as generating question and referential expression templates that use these arguments. Note that we prompt the model with additional guidelines for specific chart types, denoted by `chart_type_prompt`.

Stage 2: Extract Data and Generate Question Templates Prompt

```
You are a helpful assistant. Given a template python script as follows:
```python
{code}
```

Notice that data values are intentionally filled in with <data>
placeholders. You have access to the matplotlib fig object that this
script with the filled in data values creates.
```

```
## Your Task
Create extraction functions that extract the actual data values plotted/
displayed in the rendered matplotlib figure. These values generally
correspond to the <data> placeholders but may be approximated, rounded,
or otherwise modified during visualization.

## Steps:
1. Identify visual attributes in the script. Different attributes
represent distinct data series or groupings, such as such as individual
line series, bar groups, scatter plot series, or sets of data labels
displayed within the chart. Do not extract hardcoded values that are not
masked out with <data> placeholders – these are not the underlying data
being plotted.
2. For each attribute, create two functions following the pattern below
3. Extract data as it appears visually – approximated/binned values are
acceptable

## Data Extraction and Kwargs Generation Functions
- Name: extract_and_generate_kwargs_<attribute_name>(fig)
- Extract the actual data values for the given attribute that is plotted/
displayed in the chart. Infer the attribute name from the context of the
template script.
- DO NOT extract axis labels, legend labels, tick labels, titles, or
other chart annotations used in interpreting the chart
- DO NOT round the data values after extracting them from the matplotlib
figure
- Return a list of dictionaries, where each dictionary contains:
    – A "data" key with the extracted value (number or string only),
    which is used to format the answer template
    – Additional kwargs that capture all dimensions of variation needed
    to identify the data value, which are used to format the question
    template
- Values should match what's visually plotted, which may be approximated/
rounded from original data
- Extracted data can be either numerical values or text strings,
depending on what is plotted

## Question and Answer Template Functions
- Name: get_templates_<attribute_name>()
- Return a tuple containing (question_template, answer_template) as
strings with {{placeholder}} variables
- Questions must be answerable from the chart image alone
- Answer template should simply be "{{data}}"
- DO NOT round any formatted values in the question or the answer
template
- Use descriptive references to identify the extracted data based on
visual elements observable in the chart image itself (e.g., category
names, legend labels, axis values, colors, positions). DO NOT USE generic
 index-based references like "first bin", "second point", or "item 1",
which are ambiguous when looking at the rendered chart
- Templates should work for all data values in the attribute and yield
unique questions for each data value

## Additional Instructions
- Access legend elements: Use ax.get_legend() to get the legend object,
then legend.get_children() to access the individual legend components (
text labels, colored patches, lines).
{chart_type_prompt}
## Format:
```python
# Imports here (matplotlib.pyplot, numpy, etc.)

def extract_and_generate_kwargs_<attribute_name>(fig):
    \"\"\"
```

```
    Extract plotted data values and generate kwargs for question
    formatting
    \"\"\"
    kwargs_list = []
    # Extract data and build kwargs dictionaries
    # Each dict should include 'data' key, and other identifying
    parameters
    return kwargs_list  # List of dicts, each with 'data' and other
    kwargs

def get_templates_<attribute_name>():
    \"\"\"
    Return question and answer templates with {{placeholders}}
    \"\"\"
    question_template = "What is the {{metric}} for {{category}}?"  #
    Example
    answer_template = "{{data}}"  # Always just {{data}}
    return question_template, answer_template
```
Create separate code blocks for each visual attribute. Each function must
 be self-contained.
```

Below is the prompt for extracting bounding boxes.

### Stage 3: Extract Bounding Boxes Prompt

```
You are a helpful assistant. Given a previously created data extraction
function for a single attribute, modify it to add bounding box generation
 capabilities.

```python
{extract_functions}
```

## Task
Modify the provided extract_and_generate_kwargs_<attribute_name> function
 and get_templates_<attribute_name> function to include bounding box
functionality.

## Required Modifications:

### Modify Data Extraction Function
- Update extract_and_generate_kwargs_<attribute_name>(fig) to include "
bbox" key in returned dictionaries
- The "bbox" key refers to the bounding box as a matplotlib Bbox object
that localizes the visual element representing this data point. The
bounding box must be in display coordinates. Follow the below tips:
    - Target the most specific visual element for each data point (e.g.
    marker, bar segment, text label)
    - Use ax.get_window_extent() when possible.
    - Avoid converting to display coordinates manually when the bounding
    box can be obtained directly from the matplotlib object.
- CRITICAL: Each data value must have a unique bounding box

### Modify Template Function
- Update get_templates_<attribute_name>() to return 4-tuple: (
question_template, answer_template, bbox_question_template,
bbox_answer_template)
- Bbox question template: "Provide the bounding box coordinates of the
region this sentence describes: [description using same placeholders as
data question]"
- Bbox answer template: always "{{bbox}}"
{chart_type_prompt}
## Expected Output Format:
```python
```

```
# Imports here

def extract_and_generate_kwargs_<attribute_name>(fig):
    \"\"\"Extract plotted data values and generate kwargs with bounding
    boxes\"\"\"
    kwargs_list = []
    # Extract data and build kwargs dictionaries with 'data', 'bbox', and
     identifying parameters
    return kwargs_list

def get_templates_<attribute_name>():
    \"\"\"Return question, answer, and bbox templates\"\"\"
    question_template = "What is the {{metric}} for {{category}}?"
    answer_template = "{{data}}"
    bbox_question_template = "Provide the bounding box coordinates of the
     region this sentence describes: the {{metric}} bar for {{category}}"
    bbox_answer_template = "{{bbox}}"
    return question_template, answer_template, bbox_question_template,
    bbox_answer_template
```

Provide the complete modified functions for this specific attribute.

Table 6: Additional Instructions for Extracting Data for Each Chart Type

| Chart Type | Additional Instructions |
| --- | --- |
| Area | At any given x-coordinate in a stackplot, each layer's polygon has both a top edge and a bottom edge. The individual layer height is simply: `individual_height = y_top - y_bottom` at that x-coordinate. Follow the below approach to extract individual layer heights:
• **Extract x-coordinates**: Get the unique x-values (data points) from the polygon vertices
• **For each collection (layer) and each x-coordinate**: Find all vertices at that exact x-coordinate (within small tolerance). Get `y_top = max(y_values_at_x)` and `y_bottom = min(y_values_at_x)`. Calculate `individual_height = y_top - y_bottom`
• **Generate kwargs**: Include the individual height, layer name, and x-coordinate |
| Errorbar | • `ax.containers` will contain both BarContainer and ErrorbarContainer objects. Check the type of the container and process accordingly.
• For ErrorbarContainer objects, check `container.has_xerr` and `container.has_yerr` to determine error bar orientation.
• `container.lines` is a tuple with structure like `(data_line, (cap_line1, cap_line2), (line_collection,))`. Use `container.lines[2][0]` to get the error bar stems.
• When processing the unpacked line objects, use the `has_xerr`/`has_yerr` results to determine which coordinates contain the error information. |
| Scatter | If individual points are not uniquely identifiable by categorical labels, focus on aggregate statistics:
• Use `ax.collections` to get scatter data points, then compute aggregate values from the x,y coordinates, such as cluster means, ranges of values, etc. The bounding box template should refer to the cluster of points used to compute the aggregate statistic. |
| Density | When extracting values from a continuous distribution or function, you must first identify the tick values from `ax.get_xticks()` or `ax.get_yticks()`. Extract ONLY the data points at these tick values. Do NOT extract data at other arbitrary points. |
| Hist | • Each question should ask about the height of a specific bin within a specific interval.
• Determine intervals by obtaining the x-tick positions using `ax.get_xticks()`. For every pair of adjacent x-ticks, define an interval `[xtick_i, xtick_{i+1}]`.
• For each interval, include only bars whose centers fall within that range. You MUST refer to these bins using ordinal numbers (first, second, third, etc.) from left to right within each interval. DO NOT use generic index-based references like "bin 1", "bin 2", or "item 1", which are ambiguous when looking at the rendered chart. |

Table 7: Additional Bounding Box Generation Instructions for Each Chart Type

| Chart Type | Additional Instructions |
|---|---|
| Area | When visible markers are present, compute the bounding box around each marker. Otherwise, when the extracted values are heights of layers, compute the center of the bounding box at the top of the layer at each x-coordinate and create a square bounding box with $10 \times 10$ pixel dimensions centered at that point. Do NOT use other bounding box dimensions. |
| Line | When visible markers for the data are present, compute the bounding box around each marker. Otherwise, when there are no markers, create a square bounding box with $10 \times 10$ pixel dimensions for each data point. Do NOT use other bounding box dimensions. |
| Radar | When visible markers for the data are present, compute the bounding box around each marker. Otherwise, when there are no markers, create a square bounding box with $10 \times 10$ pixel dimensions for each data point. Do NOT use other bounding box dimensions. |
| Density | If the extracted values are the density at specific coordinates, create a square bounding box with $10 \times 10$ pixel dimensions centered at the point on the density curve corresponding to each coordinate. Do NOT use other bounding box dimensions. |
| Scatter | If the extracted values are aggregate statistics, compute the bounding box over the relevant cluster of points. If the extracted values are individual points, compute the bounding box over each point. |
| Error Bar | Manually calculate the bounding box using the segment endpoints: use `ax.transData.transform` to convert the two segment points to display coordinates. Create a Bbox from the transformed coordinates using `Bbox.from_bounds(min_x, min_y, width, height)`. For the dimension perpendicular to the error bar, use a fixed width of 5 pixels. |

## C  DATA POSTPROCESSING PROMPTS

Below is the prompt to determine the error margin for generated questions and answers.

Postprocessing: Error Margin Prompt

```
You are given a chart image and several questions about the chart.

{questions}

## Your Task:
Analyze the chart's visual elements to determine the appropriate
precision level of the ground truth answer and establish a reasonable
error margin for evaluating predicted answers. You should reason about
how you would answer these questions based on the chart image and
identify the relevant visual elements.

### Step-by-Step Analysis:

**STEP 1: Identify Answer Type**
- **Direct readings**: Values explicitly shown as text labels, legend
items, category names, or data point labels
- **Estimated values**: Values that must be visually interpolated from
axis positions, calculated, or derived

**STEP 2: For Direct Readings:**
- **Ground truth precision**: If the answer is a number, use the
precision of the number in the text label (e.g., "42.5" has precision 1)
- **Error margin**: 0 (exact match required for directly labeled values)
- **Special case**: If answer is categorical/text, set precision to null
and error margin to 0
```

**STEP 3: For Estimated Values: Determine Ground Truth Precision and Error Margin**

Look at the axis tick labels and identify the **smallest meaningful unit of difference** between consecutive ticks. The ground truth should be **one decimal place more precise** than the tick interval, while the error margin accounts for visual estimation uncertainty.

### Tick Interval Analysis Examples:

**Interval = 1 (ticks: 1, 2, 3, 4, 5)**
- Ground truth precision: 1 (tenths place) – one level more precise than interval
- Error margin: 0.5 (half the tick interval)
- Reasoning: Ground truth can be 2.3, 4.7, etc. Visual tolerance is +/-0.5

**Interval = 10 (ticks: 10, 20, 30, 40, 50)**
- Ground truth precision: 0 (ones place) – one level more precise than interval
- Error margin: 5 (half the tick interval)
- Reasoning: Ground truth can be 23, 47, etc. Visual tolerance is +/-5

**Interval = 20 (ticks: 20, 40, 60, 80, 100)**
- Ground truth precision: 0 (ones place) – one level more precise than interval
- Error margin: 10 (half the tick interval)
- Reasoning: Ground truth can be 33, 67, etc. Visual tolerance is +/-10

**Interval = 0.1 (ticks: 0.1, 0.2, 0.3, 0.4)**
- Ground truth precision: 2 (hundredths place) – one level more precise than interval
- Error margin: 0.05 (half the tick interval)
- Reasoning: Ground truth can be 0.23, 0.37, etc. Visual tolerance is +/-0.05

**Interval = 0.5 (ticks: 0.5, 1.0, 1.5, 2.0)**
- Ground truth precision: 2 (hundredths place) – one level more precise than interval
- Error margin: 0.25 (half the tick interval)
- Reasoning: Ground truth can be 1.23, 1.67, etc. Visual tolerance is +/-0.25

### Precision Scale Reference:
The precision can be **any integer** (positive, negative, or zero) or **null**:

- **null**: Categorical/text values
- **3**: Thousandths place (0.001)
- **2**: Hundredths place (0.01)
- **1**: Tenths place (0.1)
- **0**: Ones place (1)
- **-1**: Tens place (10)
- **-2**: Hundreds place (100)
- **-3**: Thousands place (1000)

### Ground Truth Precision Rule:
**For estimated values**: Ground truth precision = interval_precision + 1
**For direct readings**: Ground truth precision matches the precision shown in labels

### Complex Cases:
- **Multiple axes**: Use the axis most relevant to the answer
- **Logarithmic scales**: Focus on the linear spacing between major ticks

```
- **Mixed scales**: Choose the most restrictive precision that's
reasonable for the ground truth
- **Percentage charts**: Disregard the percentage sign, focus on
numerical values
- **Calculated values**: Consider the precision of the least precise
component

Output your analysis as a JSON object:
```json
{{
    "answer_type": "direct_reading" or "estimated_value",
    "ground_truth_precision": <integer or null>,
    "error_margin": <number or 0>,
    "tick_interval": <the interval between consecutive ticks, if
    applicable>,
    "reasoning": "Detailed explanation: Ground truth precision [X] (one
    level more precise than tick interval [Y]) because [reason]. Error
    margin [Z] (half tick interval) because [visual estimation tolerance
    reasoning]"
}}
```
```

Below is the prompt to evaluate question quality.

### Postprocessing: Question Quality Prompt

```
You are given a chart image and several questions about the chart.

{questions_section}

## Your Task:
Evaluate the overall quality and answerability of this set of questions
given the chart image. You should reason about how you would answer these
 questions based on the chart image. Since all questions follow a similar
 reasoning pattern, judge them as a group and determine which category
they fall into:

1. **VALID**: Questions that can be answered by examining the chart, even
 if they require:
   - Making reasonable assumptions about standard chart elements (axes,
   legends, data points)
   - Basic chart reading skills (identifying trends, comparing values,
   reading labels)
   - Standard domain knowledge (e.g., knowing that "Q1" means first
   quarter)
   - IMPORTANT: Questions with precise numerical parameters are valid as
   long as the underlying data is represented in the chart.

2. **AMBIGUOUS**: Questions that are genuinely unanswerable because:
   - Key terms are completely undefined AND cannot be inferred from chart
    context
   - The question refers to chart elements that definitively don't exist

3. **DEFECTIVE**: Questions where:
   - The answer is explicitly stated in the question text itself
   - No chart examination is needed because the answer is given away in
   the question

Output a JSON object with the following fields:
```json
{{
    "questions_quality": "<AMBIGUOUS|VALID|DEFECTIVE>",
    "justification": "<Brief explanation for your judgment of the
    question set>"
}}
```
```

```
```
```

# D  EXPERIMENT DETAILS

## D.1  MULTIMODAL LLM EVALUATION

In Table 8, we detail the standard, chain-of-thought, and set-of-marks prompts that we use for evaluating chart question-answering. Table 9 shows the prompt we use for evaluating visual grounding. Additionally, for GPT-4o, we instruct the model to output normalized coordinates in [0, 1]. Qwen2.5-VL natively outputs coordinates relative to the resized image, and InternVL3 is prompted to normalize coordinates by 1000.

Table 8: Chart Question Answering Prompts

| Prompt Type | Prompt |
|---|---|
| Standard | You are an expert in analyzing charts. Your task is to answer the question based on the chart provided. |
| | At the end of your response, provide your final answer in the format "Answer: X" where X is your final answer. |
| | Question:{question} |
| CoT | You are an expert in analyzing charts. Your task is to answer the question based on the chart provided. Think step by step and provide your reasoning before giving the final answer. |
| | At the end of your response, provide your final answer in the format "Answer: X" where X is your final answer. |
| | Question:{question} |
| Set-of-Marks | You are an expert in analyzing charts. Your task is to answer the question based on the chart provided. Think step by step, and in your reasoning, refer to relevant bounding boxes labeled with numbers. |
| | At the end of your response, provide your final answer in the format "Answer: X" where X is your final answer. |
| | Question: {question} |

Table 9: Visual Grounding Prompt

| Prompt Type | Template Content |
|---|---|
| Standard | You are an expert in analyzing charts. Your task is to localize the visual element corresponding to a given region. |
| | At the end of your response, provide your final answer in the format "Answer: [x1, y1, x2, y2]" where [x1, y1] are the coordinates of the top-left corner of the bounding box and [x2, y2] are the coordinates of the bottom-right corner of the bounding box. |
| | Provide the bounding box coordinates of the region this sentence describes: {referential_expression} |

## D.2  CHART ELEMENT LOCALIZATION

For object detection, we finetune 6 models: MM-GD (Zhao et al., 2024), LLMDet (Fu et al., 2025), Faster R-CNN (Ren et al., 2017), YOLOv3 (Redmon & Farhadi, 2018), RTMDet (Lyu et al., 2022), and Co-DETR Zong et al. (2023). For visual grounding, we finetune MM-GD and LLMDet.

For each model, we specify the number of epochs, batch size, and learning rate when fine-tuning on CHARTREF's training set in Table 10 shows the fine-tuning hyperparameters. The training was done on 4 NVIDIA H200s.

Table 10: Training hyperparameters

| Object Detection | | | | | | |
|---|---|---|---|---|---|---|
| Hyperparameters | MM-GD | LLMDet | Faster R-CNN | YOLOv3 | RTMDet | Co-DETR |
| Optimizer | AdamW | AdamW | SGD | SGD | AdamW | AdamW |
| $E$ | 100 | 100 | 100 | 120 | 50 | 30 |
| $\mathcal{B}$ | 4 | 16 | 4 | 4 | 16 | 16 |
| $lr$ | $1e-4$ | $1e-4$ | 0.02 | $1e-3$ | $4e-3$ | $1e-4$ |
| weight_decay | $1e-4$ | $1e-4$ | $1e-4$ | $5e-4$ | 0.05 | $1e-4$ |
| **Visual Grounding** | | | | | | |
| Optimizer | AdamW | AdamW | - | - | - | - |
| $E$ | 8 | 16 | - | - | - | - |
| $\mathcal{B}$ | - | - | - | - | - | - |
| $lr$ | $1e-4$ | $1e-4$ | - | - | - | - |
| weight_decay | $1e-4$ | $1e-4$ | - | - | - | - |

## E CHARTREF EXAMPLES

Below are visual grounding examples, where each referential expression corresponds to a red bounding box labeled with a number.

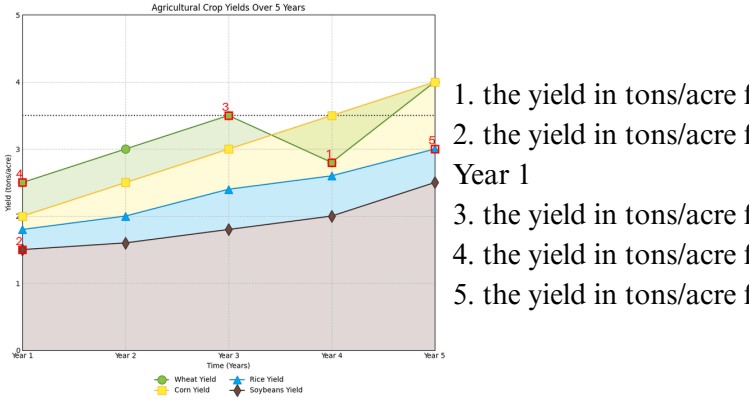

1. the yield in tons/acre for Wheat in Year 4
2. the yield in tons/acre for Soybeans in Year 1
3. the yield in tons/acre for Wheat in Year 3
4. the yield in tons/acre for Wheat in Year 1
5. the yield in tons/acre for Rice in Year 5

Figure 7: Area Chart, Example 1

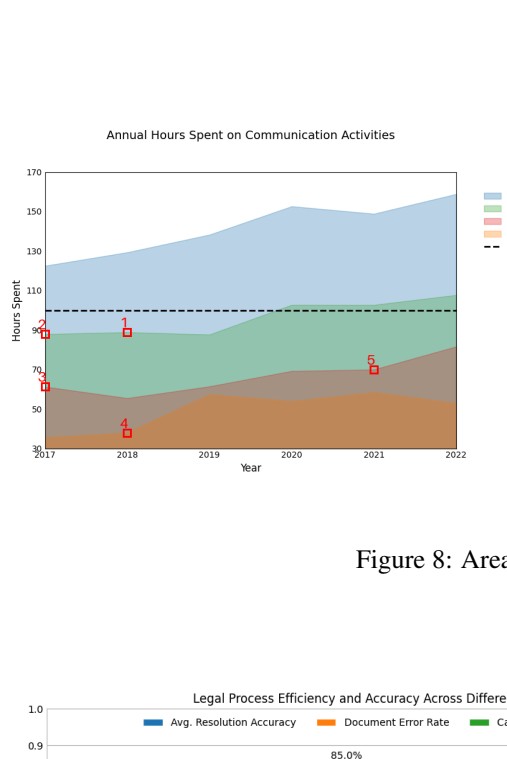

1. the annual hours spent on Meetings in 2018
2. the annual hours spent on Meetings in 2017
3. the annual hours spent on Meetings in 2017
4. the annual hours spent on Virtual Conferences in 2018
5. the annual hours spent on Calls in 2021

Figure 8: Area Chart, Example 2

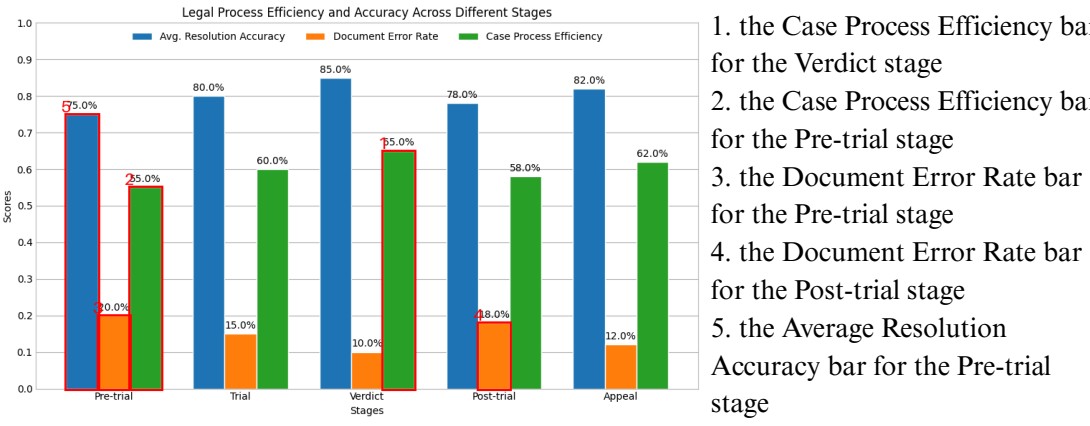

1. the Case Process Efficiency bar for the Verdict stage
2. the Case Process Efficiency bar for the Pre-trial stage
3. the Document Error Rate bar for the Pre-trial stage
4. the Document Error Rate bar for the Post-trial stage
5. the Average Resolution Accuracy bar for the Pre-trial stage

Figure 9: Bar Chart, Example 1

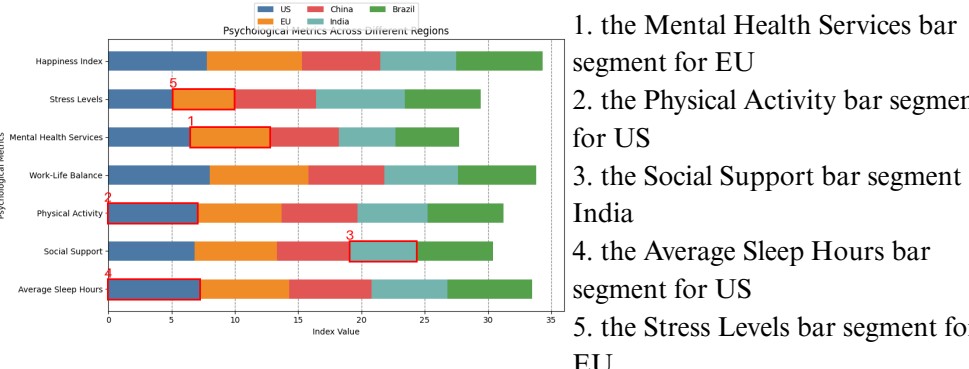

1. the Mental Health Services bar segment for EU
2. the Physical Activity bar segment for US
3. the Social Support bar segment for India
4. the Average Sleep Hours bar segment for US
5. the Stress Levels bar segment for EU

Figure 10: Bar Chart, Example 2

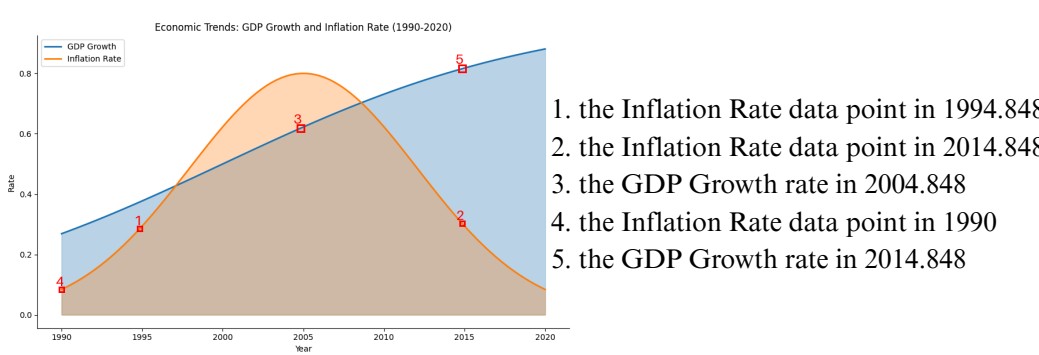

1. the Inflation Rate data point in 1994.848
2. the Inflation Rate data point in 2014.848
3. the GDP Growth rate in 2004.848
4. the Inflation Rate data point in 1990
5. the GDP Growth rate in 2014.848

Figure 11: Density Plot, Example 1

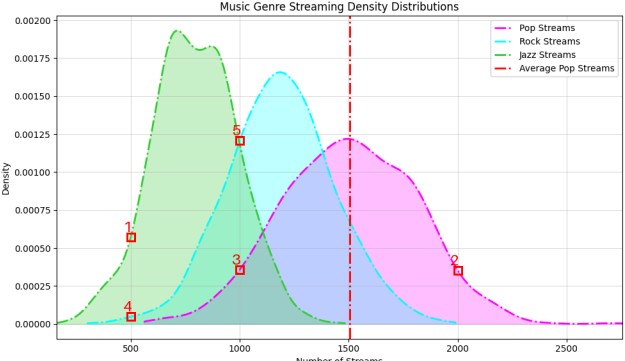

1. the density value for Jazz Streams at 500 streams
2. the density value for Pop Streams at 2500 streams
3. the density value for Pop Streams at 1000 streams
4. the density value for Rock Streams at 500 streams
5. the density value for Rock Streams at 1000 streams

Figure 12: Density Plot, Example 2

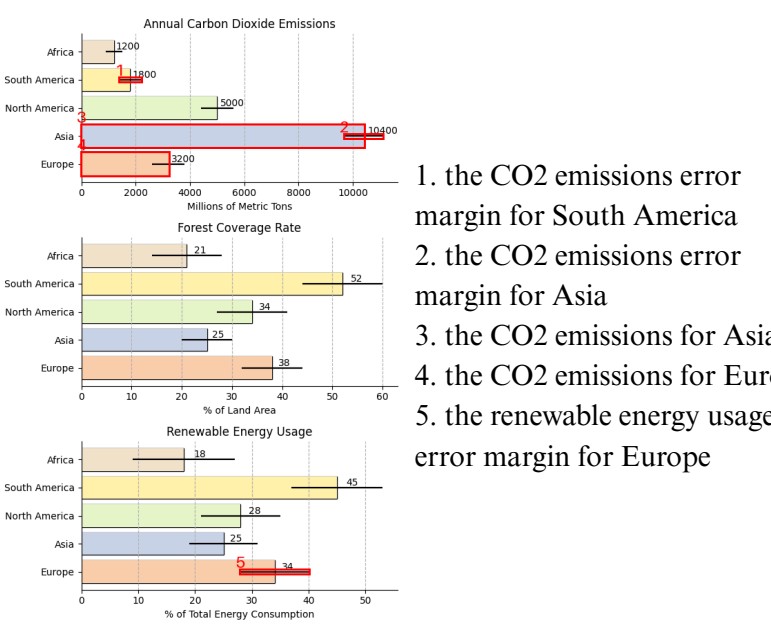

1. the CO2 emissions error margin for South America
2. the CO2 emissions error margin for Asia
3. the CO2 emissions for Asia
4. the CO2 emissions for Europe
5. the renewable energy usage error margin for Europe

Figure 13: Error Bar Chart, Example 1

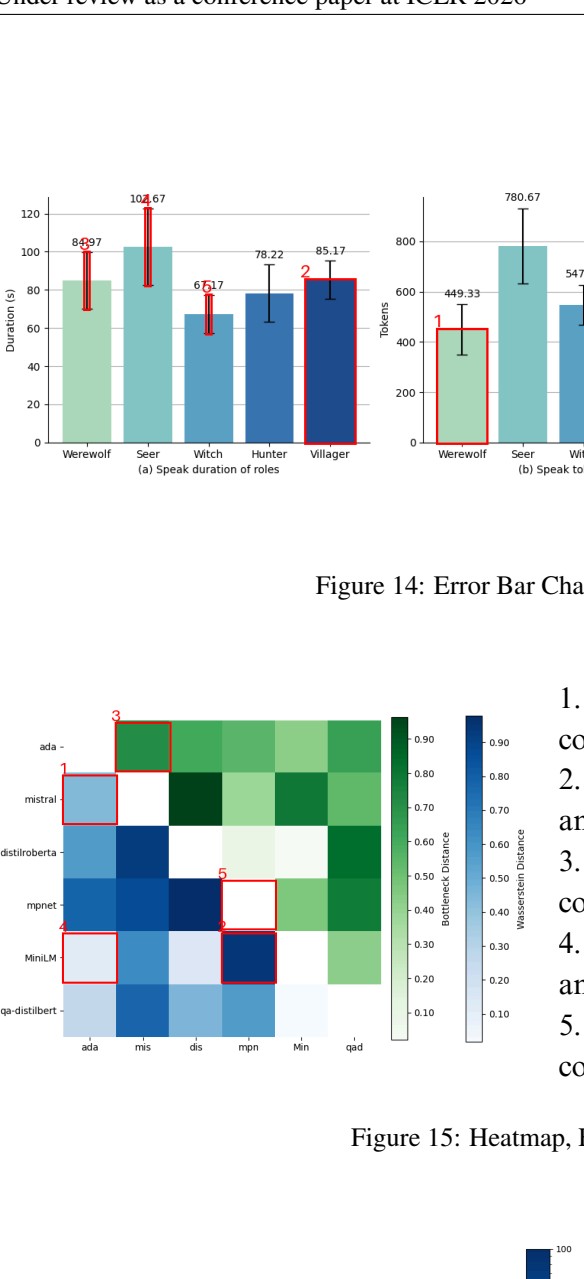

1. the mean number of speak tokens bar for the Werewolf role

2. the mean speak duration bar for the Villager role

3. the error bar for the speak duration of the Werewolf role

4. the error bar for the speak duration of the Seer role

5. the error bar for the speak duration of the Witch role

Figure 14: Error Bar Chart, Example 2

1. the heatmap cell at row 'mistral' and column 'ada'

2. the heatmap cell at row 'MiniLM' and column 'mpn'

3. the heatmap cell at row 'mpnet' and column 'mis'

4. the heatmap cell at row 'MiniLM' and column 'ada'

5. the heatmap cell at row 'mpnet' and column 'mpn'

Figure 15: Heatmap, Example 1

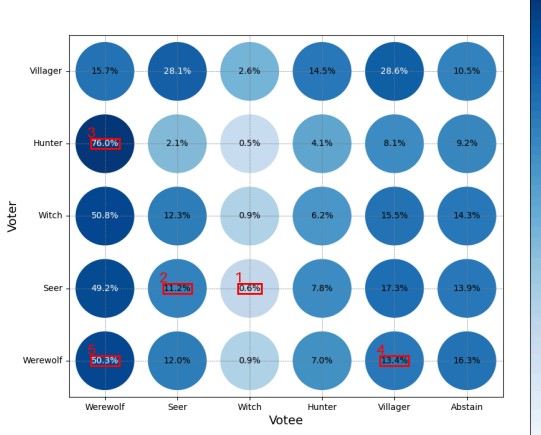

1. the percentage value showing votes Seer casts for Witch

2. the percentage value showing votes Seer casts for Seer

3. the percentage value showing votes Hunter casts for Werewolf

4. the percentage value showing votes Werewolf casts for Villager

5. the percentage value showing votes Werewolf casts for Werewolf

Figure 16: Heatmap, Example 2

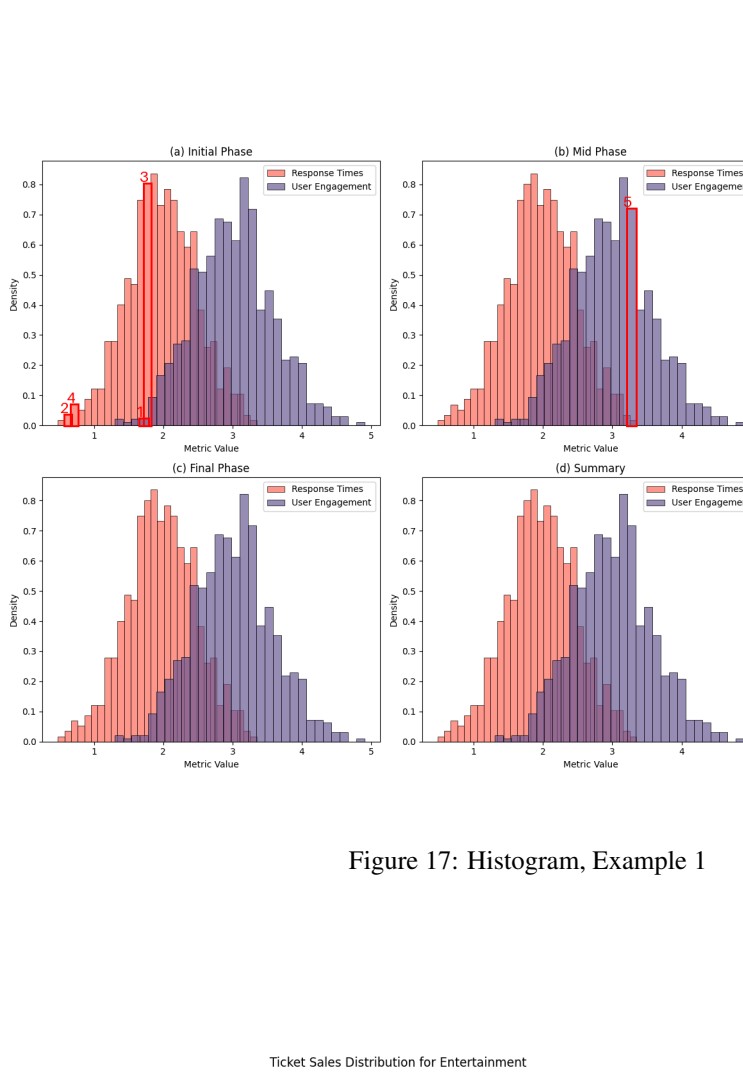

Figure 17: Histogram, Example 1

1. the fourth User Engagement bin in the interval [1, 2] in subplot (a) Initial Phase

2. the second Response Times bin in the interval [0, 1] in subplot (a) Initial Phase

3. the ninth Response Times bin in the interval [1, 2] in subplot (a) Initial Phase

4. the third Response Times bin in the interval [0, 1] in subplot (a) Initial Phase

5. the third User Engagement bin in the interval [3, 4] in subplot (b) Mid Phase

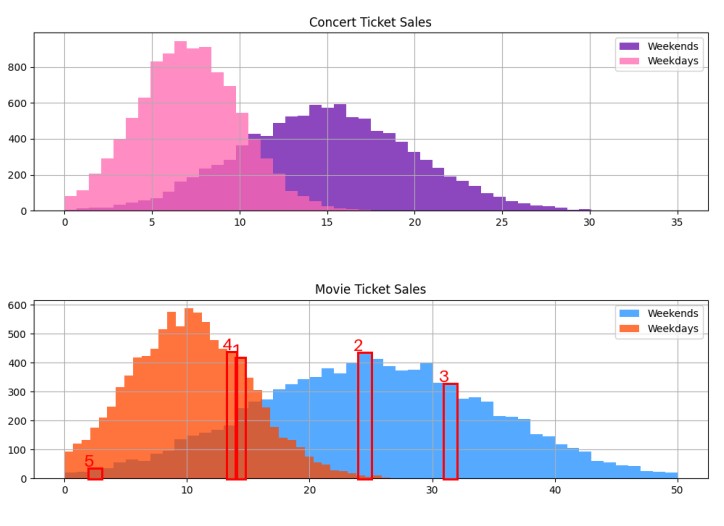

Figure 18: Histogram, Example 2

1. the seventh bar for Weekdays in the interval [10, 20) in the Movie Ticket Sales histogram

2. the fifth bar for Weekends in the interval [20, 30) in the Movie Ticket Sales histogram

3. the second bar for Weekends in the interval [30, 40) in the Movie Ticket Sales histogram

4. the sixth bar for Weekdays in the interval [10, 20) in the Movie Ticket Sales histogram

5. the first bar for Weekends in the interval [20, 30) in the Movie Ticket Sales histogram

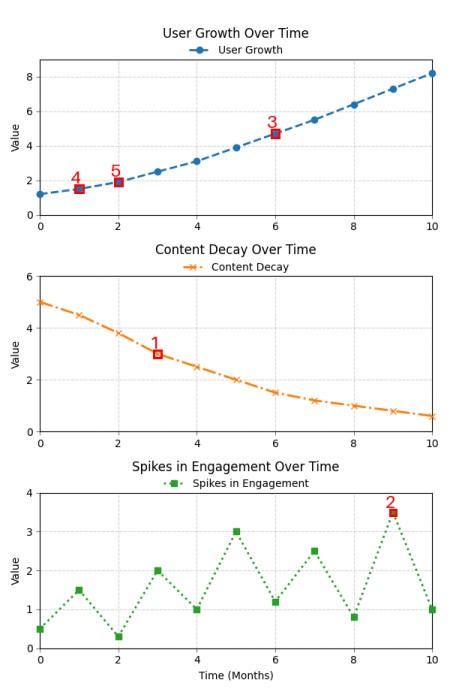

1. the content decay value at time point 3 months
2. the spikes in engagement value at time point 9 months
3. the user growth value at time point 6 months
4. the spikes in engagement value at time point 1 months
5. the user growth value at time point 2 months

Figure 19: Line Chart, Example 1

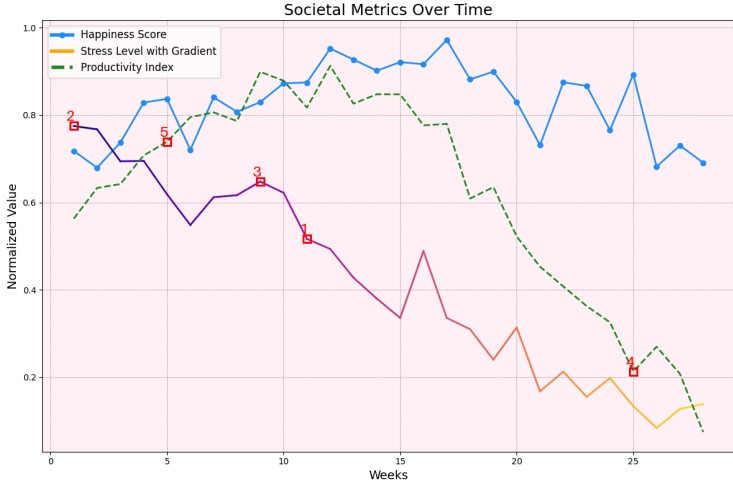

1. the Stress Level for week 11
2. the Stress Level for week 1
3. the Stress Level for week 9
4. the Productivity Index for week 25
5. the Productivity Index for week 5

Figure 20: Line Chart, Example 2

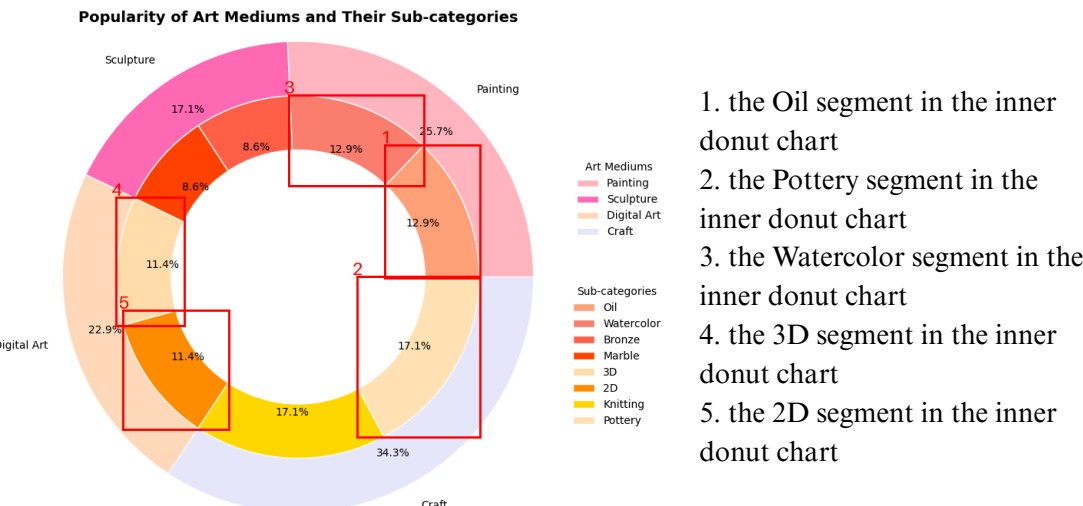

Distribution of Cases Handled by the Law Firm

1. the pie slice for Criminal cases
2. the pie slice for Family cases
3. the pie slice for Civil cases
4. the pie slice for Labor cases
5. the pie slice for IP cases

Figure 21: Pie Chart, Example 1

Popularity of Art Mediums and Their Sub-categories

1. the Oil segment in the inner donut chart
2. the Pottery segment in the inner donut chart
3. the Watercolor segment in the inner donut chart
4. the 3D segment in the inner donut chart
5. the 2D segment in the inner donut chart

Figure 22: Pie Chart, Example 2

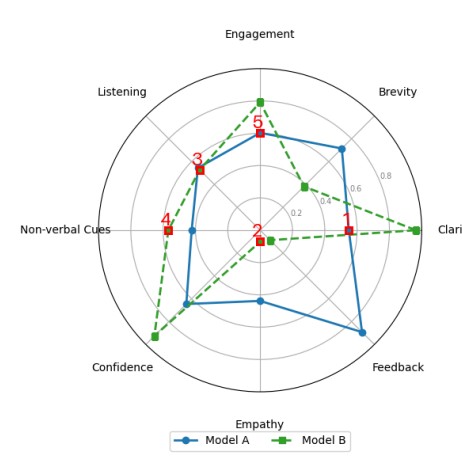

1. the Clarity score for Model A

2. the Empathy score for Model B

3. the Listening score for Model B

4. the Non-verbal Cues score for Model B

5. the Engagement score for Model A

Figure 23: Radar Chart, Example 1

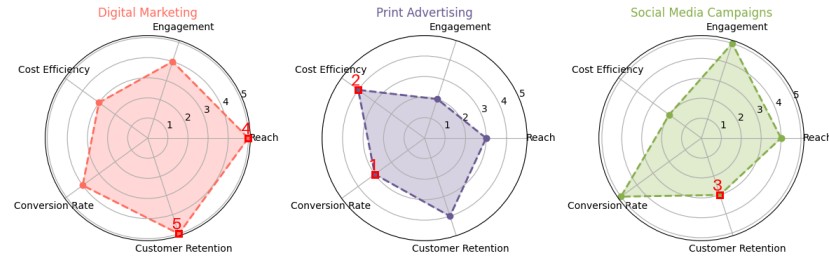

1. the Conversion Rate value for Print Advertising

2. the Cost Efficiency value for Print Advertising

3. the Customer Retention value for Social Media Campaigns

4. the Reach value for Digital Marketing

5. the Customer Retention value for Digital Marketing

Figure 24: Radar Chart, Example 2

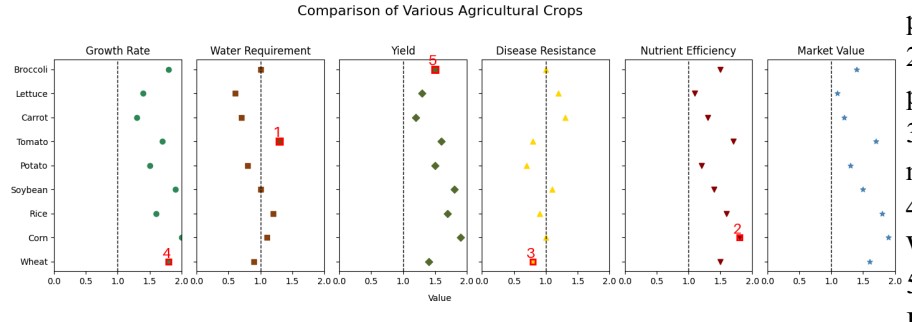

1. the Water Requirement data point for Tomato

2. the Nutrient Efficiency data point for Corn

3. the Disease Resistance marker for Wheat

4. the Growth Rate marker for Wheat

5. the Yield data point for Broccoli

Figure 25: Scatter Plot, Example 1

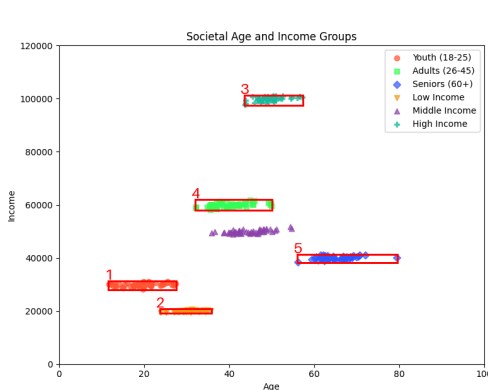

1. the mean x-coordinate (Age) of the Youth (18-25) cluster
2. the mean x-coordinate (Age) of the Low Income cluster
3. the mean y-coordinate (Income) of the High Income cluster
4. the mean y-coordinate (Income) of the Adults (26-45) cluster
5. the mean x-coordinate (Age) of the Seniors (60+) cluster

Figure 26: Scatter Plot, Example 2

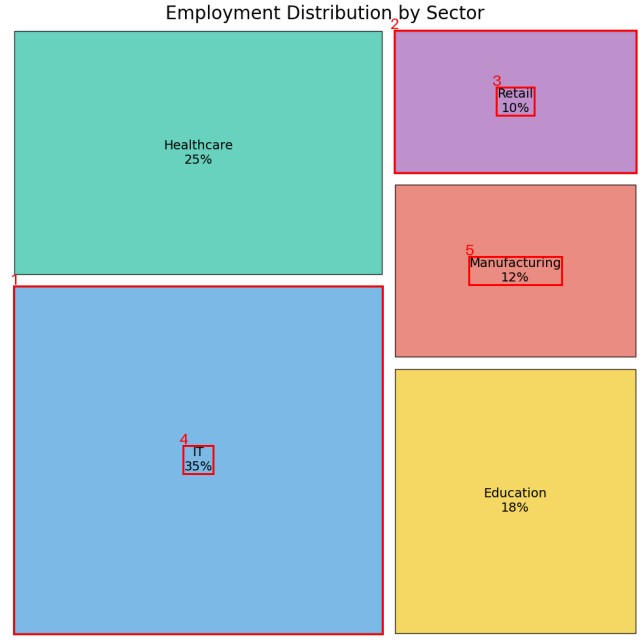

1. the IT sector in the treemap
2. the Retail sector in the treemap
3. the percentage displayed for the Retail sector in the treemap
4. the percentage displayed for the IT sector in the treemap
5. the percentage displayed for the Manufacturing sector in the treemap

Figure 27: Treemap, Example 1

Figure 28: Treemap, Example 2

1. the market share percentage for Nuclear

2. the market share percentage for Coal

3. the market share percentage for Solar

4. the market share percentage for Hydro

5. the market share percentage for Natural Gas

