# OpenReview forum: "ChartRef: Benchmarking Fine-Grained Visual Element Localization in Charts"
_ICLR.cc/2026/Conference — ICLR 2026 Conference Withdrawn Submission_

### Official Review · Reviewer_Ck79 · 2025-10-29

**Soundness:** 3
**Presentation:** 3
**Contribution:** 3
**Rating:** 6
**Confidence:** 4

**Summary:**

The paper introduces CHARTREF, a new benchmark targeting fine-grained visual element localization in charts. The core idea is to leverage the chart-rendering code (primarily matplotlib Python scripts) as a privileged signal to automatically produce aligned tuples of (question, answer, referential expression, bounding box) at scale. The pipeline prompts an LLM to: (i) mask out plotted data while preserving semantics (labels, legends); (ii) infer which plotted series correspond to which visual encodings; (iii) programmatically query the figure object to extract data values and transform them to pixel coordinates; and (iv) synthesize referential expressions and bounding boxes for the relevant marks. The resulting resource comprises 38,846 visual-grounding examples filtered across 1,141 figures and 11 chart types (with 44,345 total QA pairs and 40,336 detection annotations reported before filtering). Using CHARTREF, the authors show that supplying ground-truth bounding boxes improves chart QA accuracy by 3–7% over standard prompting; they also benchmark object detection (up to ~80.6 AP@50) and visual grounding (best 2.8 Acc@1 after finetuning), revealing a persistent gap in chart phrase grounding despite strong detection. The work argues for tighter vision–language alignment to exploit chart context cues (axes, legends, subplots) and provides prompts and stats to support reproducibility.

**Strengths:**

1. Introduces a code-centric supervision pipeline that (i) scales, (ii) preserves semantics (legend/axes roles), and (iii) yields aligned QA and localization—addressing limitations of image-only annotations.

2. Comprehensive benchmarking across closed and open VLMs and vision detectors; consistent improvements from providing bounding boxes for QA (3–7%); solid AP@50 after finetuning; careful qualitative analyses of error modes (saliency over-reliance, cluster confusion).

3. Pipeline diagrams (Figures 1–2) convey the three-stage process; tasks are cleanly split (QA vs detection vs grounding) with appropriate metrics.

4. Surfaces a clear research target: chart visual grounding remains hard (Acc@1 ≈ 2–3% after finetuning) despite strong detection, motivating better text–structure alignment (axes/legends/subplots).

**Weaknesses:**

1. Training and evaluation are centered on matplotlib renderings; generalization to in-the-wild charts (PDF scans, raster artifacts, non-Python toolchains) is not empirically established. Consider a held-out real-world test set (e.g., CharXiv/ChartMuseum excerpts) annotated with boxes to quantify transfer.

2. Heavy reliance on a single proprietary LLM (for templating, extraction code, and ambiguity filtering) risks style bias and silent failure modes. Provide manual audit statistics (e.g., 500-sample verification of bbox correctness; inter-annotator agreement on ambiguity labels).

3. Because ChartMimic scripts inform both generation and evaluation, ensure strict figure-level splits and report any template/code overlap between train/val/test, particularly when finetuning detectors.

4.Acc@K is coarse for phrases that may match multiple near-duplicate marks (e.g., overlapping points). Consider phrase-aware soft matching, normalized localization error (distance to true mark normalized by axis scale), or IoU@τ, τ∈{0.25,0.5,0.75} to tease apart near misses vs gross failures.

5. Reported totals vary across sections (e.g., 38,846 vs 44,345 examples; 1,141 vs 1,259 figures). Clarify which subsets underpin each table to avoid confusion and to support reproducibility.

**Questions:**

1. Did you perform a human audit of (i) bbox correctness and (ii) ambiguity filtering? If so, please report sample sizes and error rates; if not, can you add a small audit in the camera-ready?

2. Have you evaluated zero-shot or finetuned models on non-matplotlib charts (e.g., D3, ggplot2, Excel screenshots, arXiv PDFs)? A small transfer study, even with limited annotations—would strengthen external validity.

3. How exactly are SoM overlays rendered (box color/thickness/labeling), and do style changes affect the reported 3–7% QA gains? A robustness sweep here would help practitioners adopt the method.

4. For phrases that may refer to a set of marks (e.g., clusters), how do you define a single ground-truth box? Would a set-prediction or mask-based variant better reflect the task?

---

### Official Review · Reviewer_pZt5 · 2025-10-30

**Soundness:** 2
**Presentation:** 2
**Contribution:** 2
**Rating:** 4
**Confidence:** 3

**Summary:**

This paper introduces ChartRef, a large-scale dataset designed to benchmark fine-grained visual element localization in charts. The authors leverage the Python chart-rendering code from ChartMimic to programmatically generate aligned (question, answer, referential expression, bounding box) pairs across 11 chart types, totaling over 38k examples. The authors further evaluate a diverse range of models on the proposed benchmark, including both vision-language models and pure vision models, to assess their ability to localize fine-grained visual elements in charts.

**Strengths:**

1. The paper makes use of chart-rendering code to generate aligned bounding box annotations without manual labeling, representing an elegant and efficient procedural data synthesis strategy.
2. It presents a dataset covering 11 chart types with 38,000 examples, enabling cross-task evaluation across chart question answering, object detection, and visual grounding.

**Weaknesses:**

1. Using LLMs to generate code for chart parsing is risky: hallucinations may lead to inaccurate data extraction. Human verification and evaluation are necessary, and the overall dataset quality remains uncertain.
2. Based on the examples in Appendix E, the proposed benchmark appears overly templated and simple, focusing largely on questions about specific numeric values, which has fully explored (e.g., PlotQA). Moreover, the benchmark is built incrementally on ChartMimic, which raises questions about its significance and novelty.
3. The experiments provide limited insight. The results in Sec.4.1 are largely predictable, and the object-detection setting in Sec.4.2 (as well as Sec.4.3) seems to have limited practical value for the chart domain. The authors should reconsider the experimental design to better highlight the unique value of the proposed benchmark.

**Questions:**

See Weaknesses.

---

### Official Review · Reviewer_XwLu · 2025-10-31

**Soundness:** 2
**Presentation:** 2
**Contribution:** 2
**Rating:** 2
**Confidence:** 4

**Summary:**

The paper introduces CHARTREF, a large-scale benchmark for fine-grained visual element localization and grounding in charts. It programmatically pairs questions, answers, and referential expressions with precise bounding boxes across 11 chart types (38,846 grounded examples), generated from Python chart-rendering scripts. Evaluations cover Chart Question Answering (CQA) with and without localization signals, object detection, and visual grounding. Providing ground-truth boxes yields 3-7% CQA gains, but visual grounding remains very challenging (best finetuned Acc@1 ≈ 2.8), revealing a core limitation in current multimodal models’ alignment of language with chart structure (axes, legends, labels).

**Strengths:**

The idea of extracting bounding boxes from Python rendering code is innovative and scalable. Instead of manually annotating images or relying on computer vision models, the paper extracts coordinates directly from matplotlib figure objects.

The paper makes a case that visual grounding is fundamental to chart understanding. The observation that humans localize visual elements before reasoning is intuitive, and the finding that providing ground-truth bounding boxes improves question answering by 3-7% validates this approach.

Testing across three distinct tasks (question answering, object detection, and visual grounding) provides a thorough assessment. The contrast between high object detection performance (80+ AP@50) and low visual grounding accuracy (2.8) is particularly revealing about where current models fail..

**Weaknesses:**

While 11 chart types sounds good, the paper relies entirely on ChartMimic as source material. This means the charts are all programmatically generated with clean, consistent styling. Real-world charts from papers, reports, or presentations often have messy formatting, overlapping elements, custom styling, and imperfections that won't appear here.

The paper admits that expressions "only require extracting individual data from the chart" and don't cover comparisons or relationships between multiple data points. This is a significant limitation - phrases like "the bar for Model A" are much simpler than natural language people actually use, such as "the product with the highest sales in Q3" or "the trend showing declining performance." The benchmark may not reflect real visual grounding difficulty.

While the paper shows some qualitative examples in Figures 5 and 6, there's no systematic breakdown of error types.
Why does visual grounding fail?  Is it spatial precision?  Context integration? Ambiguity handling?

The paper doesn't compare against human performance on the localization tasks. We know humans achieve high accuracy on chart question answering, but how well do humans perform on the exact visual grounding task with the exact referential expressions used here? Without this baseline, it's hard to assess whether 2.8 accuracy represents a fundamental model limitation.

Figure 4 shows the data is "balanced" across chart types, but the middle panel reveals huge variation in questions per type. Line plots have far more examples than other types.

Section 2.2 mentions using a multimodal LLM to classify questions as ambiguous, defective, or valid, but doesn't report how many questions were filtered at each stage or provide inter-annotator agreement metrics.  What percentage of initially generated questions were rejected?

All training and testing happens on ChartMimic-derived data. There's no evaluation on held-out chart sources to assess whether models generalize beyond the specific rendering style and structure of matplotlib-generated figures.

**Questions:**

Why does chain-of-thought prompting sometimes hurt performance (Table 2) while set-of-marks helps? What does this tell us about how models process visual information?

What is human accuracy on the visual grounding task using your exact referential expressions? Can you provide a quantitative grounding error taxonomy that separates reference resolution errors from bounding-box misalignment (e.g., thin lines, no markers) ?​

Needs a quantitative analysis of failure modes for visual grounding? For example, what percentage of errors are due to: wrong chart element entirely, correct element but imprecise localization, confusing similar elements, failing to parse legends/axes?

For charts without explicit markers where 10×10 boxes are used, how often do these boxes truly cover the intended visual mark, and what is the measured impact on evaluation noise (Tables 6–7) ?​ How does box size affect detection and grounding performance? Are errors mainly about choosing the wrong element or about imprecise localization?

What fraction of questions are direct readings versus interpolated/estimated numbers under your postprocessing definitions ?​

Expand grounding to relational queries (comparisons, trends, multi-mark references)? Provide chart-structure annotations (axis/legend/linking tables) or intermediate supervision to encourage language-aware alignment?

Release evaluation tools + SoM prompts to standardize testing and promote adoption. Add a real-world subset (e.g., from papers/reports) with human-verified boxes to measure domain transfer?

---

### Official Review · Reviewer_r9Nn · 2025-11-01

**Soundness:** 2
**Presentation:** 1
**Contribution:** 2
**Rating:** 2
**Confidence:** 4

**Summary:**

This paper introduces ChartRef, a new benchmark dataset for evaluating the fine-grained visual localization capabilities of models on charts. The key contribution is a scalable data generation pipeline that uses LLMs to parse Python chart-rendering scripts. This process programmatically extracts bounding boxes for visual elements (like bars and markers) and aligns them with generated question-answer pairs and referential expressions. Using this dataset, the authors benchmark VLMs and LVLMs on three tasks: Chart Question Answering (CQA), Object Detection, Visual Grounding.

**Strengths:**

- The idea of the benchmark is interesting and important to assess at the fundamental capabilities of LVLMs on structured data.
- The data generation pipeline is technically sound.

**Weaknesses:**

- The main problem of this paper is its poor organization and clarity. While the data generation pipeline seems technically sound, the motivation for the experimental sections is not clearly stated and suffers from unclear notation. For example, the motivation for Sec 4.1 is unclear. The finding that providing ground-truth bounding boxes as visual cues can improve chart QA performance is obvious and not novel. The paper itself notes this is done using Set-of-Marks (SoM) prompting, and the reviewer cannot understand the main message the authors want to deliver.

- Also, the paper suffers from multiple missing definitions and unclear notation. It is missing crucial definitions for its experimental setup, which makes the results difficult to interpret. For example, evaluation pipeline and metrics are not well-defined. The paper never explains what "AR" (Table 3) or "Acc@N" (Table 5) represent, and model abbreviations are not explained. Table 3 uses (T), (B), and (L) without defining them. The presentation of this paper is below the acceptance threshold, and reading a paper with multiple typos and unclear notation is frustrating.

- The findings lack depth. It's obvious that the zero-shot visual grounding performance of both VLMs and LVLMs on charts is pretty low. The distribution of chart images is far from their generic training data distribution. What are the findings other than just a performance drop? Is the degradation due to the vision encoder backbone not encoding the information needed, or is it due to cross-modal misalignment between the referential expression and the visual features?

- Missing analysis of the proposed dataset. The dataset mostly relies on LLM generation. However, LLMs can make mistakes, especially in such a complex task. A human evaluation should be considered to assess the quality of the dataset (for example, its correctness and diversity).

**Questions:**

- What's the visual grounding performance of Qwen-2.5-VL after fine-tuning on the proposed dataset?

---

### Note · Authors · 2025-11-17

**Comment:**

We thank the reviewers for their constructive feedback on our work.

**Withdrawal Confirmation:**

I have read and agree with the venue's withdrawal policy on behalf of myself and my co-authors.